# Reduction-Responsive Stearyl Alcohol-Cabazitaxel Prodrug Nanoassemblies for Cancer Chemotherapy

**DOI:** 10.3390/pharmaceutics15010262

**Published:** 2023-01-12

**Authors:** Yuting Liu, Xinhui Wang, Zhe Wang, Rui Liao, Qian Qiu, Yuequan Wang, Cong Luo

**Affiliations:** Department of Pharmaceutics, Wuya College of Innovation, Shenyang Pharmaceutical University, Shenyang 110016, China

**Keywords:** cabazitaxel, disulfide bond, prodrug nanoassemblies, reduction-responsive, cancer therapy

## Abstract

Cabazitaxel (CTX) has distinct therapeutic merits for advanced and metastatic cancer. However, the present clinical formulation (Jevtana^®^) has several defects, especially for undesirable tumor-targeting and serious side effects, greatly limiting the therapeutic efficacy. Small-molecule prodrug-based nanoassemblies integrate the advantages of both prodrug strategy and nanotechnology, emerging as a promising treatment modality. Herein, disulfide bonds with different lengths were employed as linkages to elaborately synthesize three redox-sensitive stearyl alcohol (SAT)-CTX prodrug-based nanoassemblies (SAC NPs, SBC NPs and SGC NPs) for seeking optimal chemotherapeutical treatment. All the prodrug-based nanoassemblies exhibited impressive drug-loading efficiency, superior self-assembly capability and excellent colloidal stability. Interestingly, the drug release behaviors of three prodrug-nanoassemblies in the same reductive environment were different owing to tiny changes in the carbon chain length of disulfide bonds, resulting in disparate cytotoxicity effects, pharmacokinetic outcomes and in vivo antitumor efficacies. Among them, SAC NPs displayed rapid drug release, excellent cytotoxicity, long blood circulation and enhanced tumor accumulation, thus showing strong tumor inhibition in the 4T1-bearing mouse model. Our study shed light on the vital role of connecting bonds in designing high-efficiency, low-toxicity prodrug nanoassemblies.

## 1. Introduction

Malignancy remains a major health impediment all over the world [1]. Over the decades, chemotherapy based on taxane derivatives (e.g., paclitaxel, docetaxel) has emerged as an efficacious and widely used method among plentiful therapeutic options [2]. Unfortunately, the present therapeutic outcomes of chemotherapy based on taxane derivatives still fall short of expectations due to severe systemic toxicities and multidrug resistance [3,4]. As a new generation of semisynthetic taxane derivative, cabazitaxel (CTX) shows a lower affinity for P-glycoprotein and superior antitumor efficacy, holding great potential for the treatment of docetaxel-resistant metastatic breast cancer [5]. However, the present clinical formulation (e.g., Jevtana^®^) added a large amount of Tween 80 and ethanol to overcome the poor solubility of CTX, resulting in further excipient-associated toxicities, such as allergic reactions, anemia, hepatotoxicity and neurotoxicity [6]. In addition, the clinical efficacy is also compromised by the poor stability, tumor-targeting ability and pharmacokinetic properties of Jevtana^®^. Therefore, constructing an emerging drug delivery system is crucial for the enhanced clinical therapeutic efficacy of CTX.

For years, nano delivery systems (nano-DDSs) have played crucial roles in delivering chemotherapeutics due to desirable tumor-targeting ability and prolonged circulation time [7,8,9,10,11,12,13,14]. However, conventional carrier-based nano-formulations usually load drugs with non-covalent interactions, leading to a series of shortcomings, such as low drug loading (usually less than 10%, *w*/*w*), potential drug leakage and adverse material-related toxicity [11,15,16,17]. Formulating drugs into prodrugs by introducing covalent and coordinate bonds holds great promise to improve drug loading and self-assembly capabilities [18,19]. Recently, prodrug-based nano-DDS have integrated the advantages of both prodrug strategy and biomedical nanotechnology, attracting tremendous scientific interest [20,21]. Particularly, small-molecule prodrug-based nanoassemblies, as both carriers and cargos, are characterized by distinct high drug-loading (>50%, *w*/*w*), accurate dosages, superior self-assembly properties and reduced excipient-related toxicity, showing more potential in cancer chemotherapy [22,23].

To better achieve on-demand cargo release at tumor sites, smart small-molecule prodrug-based nanoassemblies with internal stimuli-responsive triggers are widely developed [11,24,25]. For instance, we previously lucubrated the impact of disulfide bond bridge on paclitaxel (PTX)-oleic acid prodrug nanoassemblies and found that disulfide linkages could achieve effective reductive-selective drug release in response to excessive intracellular reductive glutathione (GSH) at tumor sites, greatly alleviating the side effects and improving antineoplastic efficacies [26]. Except for selective responsiveness, disulfide bonds also contribute to the self-assembly of prodrug-based nano-DDS by introducing molecular “structure defects” [27,28]. What is more, the distance between the disulfide bond and the ester bond is a pivotal point that has a great influence on drug release, cytotoxicity and antitumor efficacy in vivo [29,30]. Inspired by these findings, we hypothesized that there would be an optimal carbon chain length to trigger the high therapeutic effect of CTX.

Herein, we rationally selected CTX as a model drug and stearyl alcohol (SAL) as the modifier chain, synthesizing three SAL-CTX prodrug nanoassemblies (SAC NPs, SBC NPs and SGC NPs) based on disulfide bonds with different lengths (Figure 1). Then, we paid great attention to the influence of disulfide bonds at different positions on the self-assembly, colloidal stability, drug release and cytotoxicity of NPs. Finally, the antitumor efficacy of nanoassemblies was evaluated in a 4T1 xenograft mouse model, and optimal prodrug nanoassemblies (SAC NPs) with high efficiency were pointed out.

## 2. Materials and Methods

### 2.1. Material

CTX, Dithiothreitol (DTT), coumarin-6, Hoechst 33342, 3-(4,5-dimethylthiazol-2-yl)-2,5-diphenyl tetrazolium bromide (MTT) and cell culture media were purchased from Meilun Biotech Co. Ltd. (Dalian, China). Stearyl alcohol (SAT), dithiodiglycolic acid, 3,3′-dithiodibutyric acid, 4,4′-dithiodibutyric acid, 1-Ethyl-3-(3-dimethyllaminopropyl) carbodiimide hydrochloride (EDCI), hydroxybenzotriazole (HOBt) and 4-dimethylaminopyrideine (DMAP) were obtained from Aladdin, (Shanghai, China). Sterile cell culture dishes and centrifuge tubes were bought from NEST Biotechnology (Wuxi, China). 1,2-distearoyl-sn-glycero-3-phosphoethanolamine-N-[methoxy(polyethyleneglycol)-2000] (DSPE-PEG_2K_) was purchased from AVT (Shanghai) Pharmaceutical Tech Co. Ltd. (Shanghai, China). All other reagents and solvents mentioned in this article were of analytical grade.

### 2.2. Synthesis of CTX-SAT Prodrugs

Three disulfide bond-linked CTX-SAT prodrugs (SAC, SBC and SGC) were synthesized in three steps in a similar way. First, diacid (dithiodiglycolic acid, 3,3′-dithiodipropionic acid or 4,4′-dithiodibutyric acid, 1 mmol) was added to a round-bottom flask and dissolved in 3 mL acetic anhydride. Under nitrogen protection, the mixture was stirred at 25 °C for 2 h. The unreacted anhydride was then removed by spin steaming process. Then, SAT (1 mmol), DMAP (0.1 mol) and anhydrous dichloromethane were rapidly added into the above round-bottom flask, following stirring for 12 h to obtain the intermediate product. Of note, DMAP dissolved in methylene chloride beforehand, and the solution was added drop-by-drop. Silica gel column chromatography was employed to separate and purify the intermediate product. Further, the mixture solution of HOBt (1 mmol), EDCI (1.5 mmol) and DMAP (0.1 mmol) was slowly dropped in the anhydrous dichloromethane solution of the intermediate product (1 mmol) and stirred in an ice-bath for 1 h under nitrogen protection. After being brought to room temperature, CTX (1 mmol) was added, and the mixture was further stirred for 48 h under a nitrogen atmosphere. Then, dichloromethane was removed by rotary evaporation, and the final product was obtained by preparative liquid chromatography. The purified product was a white solid with a yield of 65%. The final product was confirmed by mass spectrometry and nuclear magnetic resonance spectroscopy.

### 2.3. Preparation and Characterization of Prodrug Nanoassemblies

Self-assembled prodrug nanoparticles were prepared by a one-step nanoprecipitation method. For PEGylated prodrug nanoassemblies, DSPE-PEG_2k_ (0.67 mg) and the prodrug (2 mg) were dissolved together in 0.2 mL anhydrous ethanol. Then, they were added to 2 mL of deionized water drop by drop under agitation. Upon stirring for 3 min, anhydrous ethanol in the preparation was removed by rotary evaporation at 30 °C. Eventually, deionized water was added to the nano-formulation until the final volume of the formulation was 2 mL. The non-PEGylated prodrug nanoassemblies without DSPE-PEG_2K_ were prepared by the same method.

To prepare the coumarin-6 (C6) solution, 1 mg C6 was resolved in 2 mL of anhydrous ethanol. The preparation method of C6-labeled prodrug nanoassemblies is as follows. First, the 25 μg/mL C6 ethanol solution was prepared. Then, 2 mg prodrugs (SAC, SBC or SGC) and 0.67 mg DSPE-PEG2K were dissolved with 200 μL the above C6 ethanol solution and slowly dropped into 2 mL deionized water. Finally, upon stirring for 3 min, the anhydrous ethanol in the preparation was removed by rotary evaporation at 30 °C.

Nano ZS Zetasizer (Malvern, UK) was employed to measure the particle size and Zeta potential of prodrug nanoassemblies. In addition, transmission electron microscopy (TEM, Hitachi, HT7700, Japan) was utilized to observe the morphology of the prodrug nanoassemblies.

### 2.4. Colloidal Stability, Assembly Mechanism and Molecular Docking

The colloidal stability was investigated by observing the change in the particle size of PEGylated nanoassemblies in PBS containing 10% fetal bovine serum (FBS). Briefly, 1 mg PEGylated prodrug nanoassemblies was added into 10 mL PBS containing 10% FBS. Then, the mixtures were incubated at 37 °C for 24 h with gentle shaking (*n* = 3 for each group). The particle size changes of PEGylated nanoassemblies were measured at prescriptive intervals (0, 2, 4, 8, 12 and 24 h) by Zetasizer Nano ZS (Malvern, UK).

The assembly mechanisms of SAP, SBP and SGP molecules were studied by treating prodrug nanoassemblies with 0.4 mM of NaCl, SDS or urea. Intermolecular interactions were analyzed by comparing the changes in particle size.

Molecular docking was employed to investigate the interactions between small molecules. The stable optimized structures were obtained by calculation, and the three systems were molecularly docked using the AutoDock program to obtain the combined energies.

### 2.5. In Vitro Drug Release

The in vitro drug release behavior of prodrug nanoassemblies was studied by high-performance liquid chromatography (HPLC). Specifically, 30 mL PBS containing 30% ethanol (*v*/*v*) with DTT (1, 5 and 10 mM) was employed as the release medium. The mixture without DTT was regarded as the blank release medium. Upon adding prodrug nano-formulation (200 nmol/mL, equal to CTX), the mixtures were incubated at 37 °C for 24 h with gentle shaking (*n* = 3 for each group). At the prescriptive intervals (0, 1, 2, 4, 6, 8, 12 and 24 h), 0.2 mL sample solution was collected, and the concentration of the released CTX from prodrug nanoassemblies was analyzed by HPLC (λ = 227 nm, mobile phase: acetonitrile/water = 70:30, *v*/*v*, injection volume: 30 μL).

### 2.6. Cell Culture

4T1 cells and 3T3 cells were obtained from the cell bank of the Chinese Academy of Sciences (Beijing, China). 4T1 cells were cultured in RPMI 1640 medium consisting of 10% FBS, penicillin (100 units/mL), and streptomycin (100 units/mL). 3T3 cells were cultured in a DMEM medium containing the same substances described above. All cells were incubated in a 5% CO_2_, 37 °C cell incubator.

### 2.7. Cytotoxicity

The antiproliferative activities of the CTX solution and three prodrug nanoassemblies on 4T1 cells and 3T3 cells in vitro were assessed by MTT assay. Briefly, cells were seeded into 96-well plates (2000 cells per well) and incubated for 12 h in a cell incubator at 37 °C. Then, cells were treated with serially diluted prodrug nanoassemblies and CTX solution. Upon incubating for an additional 48 h, 25 µL of the MTT solution (5 mg/mL) was added to each well and incubated for another 4 h. Then, the mixture in the wells was removed, and 200 µL dimethyl sulfoxide (DMSO) was added. After shaking for 10 min, the absorbance at 490 nm was measured by a microplate reader (ThermoFisher Scientific, Waltham, MA, USA). The GraphPad Prism 9.0 was employed to calculate IC_50_ values.

### 2.8. Cellular Uptake

The cellular uptake behavior was performed via 4T1 cells. Briefly, 4T1 cells were seeded at a density of 1 × 10^5^ cells per well into 24-well plates containing 1 mL of medium in each well and cultured for 12 h. Upon discarding the original media, 1 mL fresh culture media containing C6 solution or C6-labeled prodrug nanoassemblies (the equivalent concentration of C6 is 25 µg/mL) were added. After incubation for another 0.5 or 2 h, cells were washed three times with cold PBS, fixed with 4% formaldehyde and counterstained cell nucleus by Hoechst 33342. Finally, the PBS was employed again to wash excess C6. The fluorescence signals were observed by confocal laser scanning microscopy (CLSM, C2, Nikon, Japan).

For quantitative analysis, 4T1 cells were seeded at a density of 1 × 10^5^ cells per well into 12-well plates for 24 h. Upon discarding the original culture media, free medium-containing C6 solution or C6-labeled prodrug nanoassemblies (the equivalent concentration of C6 is 25 µg/mL) were added to each well. After incubation for another 0.5 or 2 h, the cells were washed with cold PBS. Then, the C6-containing medium was discarded, and the cells were digested by 300 µL trypsin and 1 mL media to terminate digestion. Finally, the cells were collected and re-suspended in PBS. The fluorescence signals were measured by FACSCalibur flow cytometer (BD, East Rutherford, NJ, USA).

### 2.9. Animal Studies

All animals were obtained from the Animal Centre of Shenyang Pharmaceutical University (Shenyang, China). The use of animals is approved by the Animal Ethics Committee of Shenyang Pharmaceutical University (No. 19169).

### 2.10. In Vivo Pharmacokinetic Study

The in vivo pharmacokinetic profiles of prodrug nanoassemblies with different amounts of PEG were measured by using mature male SD rats (180–220 g). First, SD rats were randomly assigned into six treatment groups (*n* = 5). After an overnight fast, the rats were intravenously administered with DiR-labled SBG NPs with 5%, 10%, 15%, 20%, 25% and 30% PEG (W_PEG_/W_Prodrug + PEG_) at an equivalent CTX dosage of 5 mg/kg. At the prescriptive interval (0, 0.087, 0.25, 0.5, 1, 2, 4, 8 and 12 h), 0.5 mL blood samples were collected and then centrifuged to get plasma. The protein precipitation method was utilized to extract prodrugs and release CTX in plasma. Finally, the concentration of prodrugs and CTX were measured by a microplate reader (ThermoFisher Scientific, Waltham, MA, USA).

The in vivo pharmacokinetic profiles of the CTX solution and prodrug nanoassemblies were measured by the following method. First, SD rats were randomly assigned into four treatment groups (*n* = 5). After an overnight fast, the rats were intravenously administered with CTX solution or three different prodrug nanoassemblies at an equivalent CTX dosage of 5 mg/kg. At the prescriptive interval (0, 0.087, 0.25, 0.5, 1, 2, 4, 8 and 12 h), 0.5 mL blood samples were collected and then centrifuged to get plasma. The protein precipitation method was utilized to extract prodrugs and release CTX in plasma. Finally, the concentration of prodrugs and CTX were analyzed by UPLC-MS/MS (Waters Co. Ltd., Milford, MA, USA).

### 2.11. Biodistribution

In vivo biodistribution study was carried out on a 4T1 tumor-bearing BALB/c mice model. First, 5 × 10^6^ of 4T1 cells suspended with 100 μL of PBS were injected subcutaneously into the right rear of BALB/c mice. When the tumor volume grew to about 300 mm^3^, the mice were treated with DiR solution and DiR-labeled prodrug nanoassemblies (1.5 mg/kg, DiR equivalent). At 24 h post-injection, the mice were killed to harvest tumors and major organs (heart, liver, spleen, lung and kidney) for fluorescence imaging by the IVIS spectrum imaging system.

### 2.12. Hemolysis Test

First, 2% RBC suspension was configured. Specifically, 10 mL of fresh rat blood was taken and stirred to remove fibrin. Then, 50 mL of normal saline was added and centrifuged at 1500 r/min for 15 min. After the supernatant was removed, the red blood cells were washed with normal saline 3 times until no red color was observed. The obtained red blood cells were prepared into a 2% suspension with normal saline. Next, 6 clean test tubes (1–4 as test subjects, 5 as positive control and 6 as negative control) were successively added to 1.5 mL 2% red blood cell suspension, 300 µL SAC NPs, 306 µL SBC NPs, 312 µL SGC NPs and 200 µL CTX solution. A total of 0.5 mL of normal saline and 0.5 mL of deionized water was used. The 6 tubes were filled with normal saline to 2 mL and incubated in a 37 °C shaker for 2 h. Finally, the supernatant was acquired and injected into the 96-well plate. The absorbance was measured at the wavelength of 545 nm.
Hemolysis rate (%) = (sample absorption − negative control absorption)/(positive control absorption − negative control absorption) × 100%

### 2.13. In Vivo Antitumor Efficacy

To investigate the in vivo antitumor efficacy of prodrug nanoassemblies, 4T1 breast tumor models were established. Upon the tumor volume reaching approximately 120 mm^3^, the mice were randomly divided into five groups (*n* = 5). Then, the mice were treated with PBS, 5 mg/kg CTX solution and SAP NPs, SBP NPs and SGP NPs at an equivalent CTX dose five times every third day via their tail vein. Tumor volume and body weight were recorded and calculated every day. The mice were sacrificed at 11 d. Meanwhile, blood samples were collected and centrifugated at 4000 rpm for 10 min for hepatic and renal function analysis. The tumors, together with major organs (heart, liver, spleen, lung and kidney), were harvested and sent to H&E analysis.

## 3. Results and Discussions

### 3.1. Synthesis of SAC, SBC and SGC

To explore the role of linkers in prodrugs, three types of reduction-responsive disulfide linkers (α-/β-/γ-position disulfide bonds) were integrated into CTX-SAL prodrugs. The three novel prodrugs were abbreviated as SAC, SBC and SGC, respectively, in accordance with the different positions of disulfide linkers on the ester bonds. The synthesis routes of the three disulfide-linked CTX-SS-SAL prodrugs are exhibited in Appendix A. The chemical structures of three prodrugs were confirmed by ^1^H NMR, ^13^C NMR and MS (Appendix A). The purity of all the three prodrugs was over 98%.

### 3.2. Preparation and Characterization of CTX Prodrug NPs

The hydrophobic CTX prodrug-based nano-formulations were prepared by a simple one-step nanoprecipitation method. At the concentrations of 100 and 200 μg/mL, homogeneous nanoparticles (NPs) could be fabricated in an ethanol–water mixed solvent system by SAC, SBC and SGC prodrugs. To further increase the solubility and stability of the prodrugs, a small amount of DSPE-PEG_2k_ was generally chosen to neutralize the high surface charge and improve the colloid stability of NPs (Figure 1A) [31]. In this work, a pharmacokinetic experiment was implemented to screen out the optimal amount of PEG. By comparing the effects of different PEG levels (5%, 10%, 15%, 20%, 25% and 30%, *w*/*w*) on in vivo fate of NPs in blood circulation, 25% PEG (*w*/*w*) was chosen to carry out the following experiments due to the longest blood circulation time of NPs containing 25% PEG (*w*/*w*) (Appendix A). As shown in Appendix A, three types of PEGylated NPs showed blue opalescence. Moreover, the spherical structures of NPs under TEM were uniform with sub-100 nm sizes, which is conducive to tumor-targeted accumulation via the high permeability and retention (EPR) effect of solid tumors (Figure 1B–D and Appendix A) [32,33,34]. In addition, slightly negative surface charges were attained for all the PEG-modified NPs (i.e., −16.40 ± 2.11 for SAC NPs, −18.63 ± 0.31 mV for SBC NPs and −21.6 ± 2.06 mV for SGC NPs, Appendix A), which would effectively prevent the aggregation of ions and improve the colloidal stability of NPs. Of note, the prodrug-based nanoassemblies acted as both carriers and cargos, contributing to the extreme drug-loading efficiency (i.e., 50.09% for SAC NPs, *w*/*w*), significantly higher than conventional nano-formulations (typically less than 10%, *w*/*w*) [22,27,35]. Moreover, the prepared nanoparticles only added a small amount of DSPE-PEG_2k_ without Tween 80 or ethanol, reducing undesirable hypersensitivity reactions of Jevtana^®^.

Then, the assembly mechanisms were explored. As shown in Figure 1E–G, the main driving force for the self-assembly process of SAC, SBC and SGC were hydrophobic interactions (grey dashed line). To further figure out the main interactions and forces during the assembly process, the nanoparticles were incubated with NaCl (0.4 M, blood concentration: 100 mM), urea (0.4 M) and sodium dodecyl sulfate (0.4 M) for 6 h at 37 °C. NaCl, urea and SDS could specifically break electrostatic interactions, hydrogen bonding and hydrophobic interactions, respectively. As shown in 1H, the particle sizes of SAC NPs, SBC NPs and SGC NPs were increased significantly in the presence of SDS, indicating that hydrophobic force played a key role in the assembly process.

Eventually, the colloidal stability of the PEGylated prodrug nanoassemblies was investigated. Three types of prodrug-based nanoassemblies exhibited excellent colloidal stability with negligible variations in diameters upon incubation in 10% FBS-supplemented PBS (pH 7.4) for 12 h at 37 °C (Figure 1I). In addition, the long-term storage stability of PEGylated prodrug nanoassemblies at 4 °C was also studied. As shown in Figure 1J, the particle size of the three NPs did not change significantly within 90 days, indicating favorable storage stability at low temperatures.

### 3.3. Reduction-Triggered Drug Release

The excellent efficacy highly depends on the selective and effective drug release from prodrug nanoassemblies in tumor cells. As displayed in Figure 2A, the three prodrugs barely hydrolyzed without DTT; however, when DTT was added, SAC NPs released nearly 80% of CTX within 4 h. Compared with SAC NPs, SBC NPs and SGC NPs released more slowly, and only about 20% of CTX could be released under 10 mM DTT at 12 h (Figure 2B–D). These results indicated that SAC NPs possessed better reduction sensitivity and could release CTX faster under tumor reduction conditions, which was conducive to playing better roles in drug therapy. The reduction-sensitive mechanism of the disulfide bonds has been clarified in our previous work [26,36]. As shown in Figure 2E, disulfide bonds were broken in the presence of dithiothreitol (DTT, GSH analog), which triggered the hydrolysis of adjacent ester bonds and the release of CTX in succession. Among them, SAC owned the shortest carbon chain, showing the fastest drug release. Despite the longest carbon chain between the disulfide bond and the ester bond of SGC, the thiol group at the γ position could attack the carbon atom on the carbonyl group to change the ester bond into a stable five-member ring thioacetone, thus promoting the release of CTX.

### 3.4. Cellular Uptake and Cytotoxicity

The cellular uptake of SAC NPs, SBC NPs and SGC NPs was evaluated by confocal microscopy and flow cytometry. As shown in Figure 3A–C, prodrug nanoassembly-treated groups possessed much stronger intracellular fluorescence than the C6 solution at both 0.5 and 2 h. In addition, given the similar particle size and surface energy, SAC NPs, SBC NPs and SGC NPs exhibited comparable intracellular uptake effects.

The cytotoxicity of NPs and the CTX solution in tumor cells (4T1 cells) and normal cells (3T3 cells) was evaluated by MTT assay. As shown in Figure 3D, the cytotoxicity of the CTX solution displayed more potent cytotoxicity than NPs on 4T1 cells, following the order of SAC NPs > SGC NPs > SBC NPs, which agreed well with drug release results. However, prodrug nanoassemblies showed much lower cytotoxicity than the CTX solution on 3T3 cells (Figure 3E and Appendix A). The tumor selective index (SI) was recorded in Appendix A. Among them, prodrug nanoassemblies possessed higher SI values than the CTX solution, especially for the SAC NPs and SGC NPs, indicating high tumor selectivity and good safety of prodrug nanoassemblies.

### 3.5. In Vivo Pharmacokinetics and Biodistribution

In order to effectively accumulate at the tumor site, NPs are supposed to maintain stability in vivo. As shown in Figure 4A, the CTX in the CTX solution was rapidly cleared from blood circulation. In comparison, PEG-modified prodrug nanoassemblies could dramatically prolong the blood circulation time of CTX. Moreover, the AUC of prodrugs in SAC NPs, SBC NPs and SGC NPs groups were approximately 39.84-, 18.30- and 9.88-times higher than CTX solution group, respectively (Appendix A), indicating the improved pharmacokinetic behavior of NPs.

Next, fluorescence imaging was applied to explore the tissue distribution and tumor accumulation of prodrug nanoassemblies. As shown in Figure 4B,C, the DiR solution mainly displayed a high fluorescent signal in the spleens, with negligible fluorescence in tumor tissues. In contrast, the fluorescence of DiR-labeled NPs, especially for SAC NPs, was increased significantly in tumor sections at 24 h, which was attributed to the long blood circulation time of PEG modification and improved tumor accumulation mediated by the EPR effect. In addition, the distribution of NPs in vivo was consistent with pharmacokinetic behavior: SAC NPs showed excellent tumor accumulation and pharmacokinetic behavior.

### 3.6. In Vivo Antitumor Efficacy

The in vivo antitumor efficacy of SAC NPs, SBC NPs and SGC NPs was studied in 4T1 tumor-bearing mice (Figure 5A). As shown in Figure 5B and Appendix A, the rapid tumor growth of the PBS group could be observed during 11 d. By contrast, varied tumor inhibition efficiencies were achieved in other treatments. Among them, the CTX solution displayed unsatisfactory antitumor activity, owing to the rapid clearance from the body post-injection and insufficient tumor accumulation. Despite slightly longer circulation time and better tumor accumulation of SBC NPs and SGC NPs, they still showed inferior antitumor activity due to the slow drug release and weak cytotoxicity. Notably, SAC NPs demonstrated potent antitumor activity, which was attributed to a higher tumor accumulation and faster drug release. Moreover, compared with the CTX solution, SAC NPs, SBC NPs and SGC NPs revealed excellent safety during treatment, with negligible change in the body weights of mice (Figure 5C). It was attributed to multiple pharmacologic advantages of prodrug nanoassemblies, including high drug loading, long circulation time and high tumor accumulation. Further, no significant hemolysis phenomenon, abnormal hepatorenal function and no obvious damage to the major organs (heart, liver, spleen, lung and kidney) were found in the NP groups (Figure 5D,E, Appendix A, Appendix A). Notably, distinct necrotic and apoptotic regions were observed in tumor tissues treated with SAC NPs, suggesting potent antitumor efficacy (Figure 5F).

## 4. Conclusions

In this study, we rationally designed three types of reduction-responsive SAL-CTX prodrug nanoassemblies (SAC NPs, SBC NPs and SGC NPs) to enhance antitumor efficiency and reduce side effects. By varying disulfide bonds with different lengths, the differences in drug release properties, cytotoxicity outcomes, blood circulation time and antitumor efficacy of three NPs were pointed out. Among them, SAC NPs with the shortest disulfide bond as the most sensitive to the reductive tumor microenvironment and was broken fastest in both the DTT solution and tumor cells, thus showing excellent drug release behaviors and cytotoxicity. In addition, SAC NPs significantly improved the AUC of CTX and exhibited better tumor accumulation capacity. As a result, SAC NPs with enhanced antitumor efficacy in a 4T1 xenograft mouse model were highlighted. Collectively, we provide an intelligent and effective strategy for the delivery of hydrophobic anticancer drugs.

## Data Availability

Not applicable.

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
