# Peer review of "Reduction-Responsive Stearyl Alcohol-Cabazitaxel Prodrug Nanoassemblies for Cancer Chemotherapy"

_pharmaceutics, 2023, doi:10.3390/pharmaceutics15010262_

Round 1

Reviewer 1 Report

In this study, the authors presented their findings on tumor treatment using three types of reduction-responsive stearyl alcohol-Cabazitaxel prodrug nanoassemblies. Although the proposed tumor growth inhibition efficacy of prodrug nanoassemblies is excellent, results and interpretations to substantiate it are lacking. Therefore, the following information should be supplemented.

1.     In Figure 1b-d, please clarify the size of the scale bar in the nanoparticle electron micrograph.

2.     In Figure 3a-c, please supplement the description of the C6 solution and the manufacturing method and principle of each nanoparticle using the C6 solution.

3.     In Figure 3d,e, the authors mentioned that "prodrug nanoassemblies showed much lower cytotoxicity than CTX solution on 3T3 cells, indicating high tumor selectivity and good safety of prodrug nanoassemblies". The prodrug nanoassemblies show low cytotoxicity in both 4T1 and 3T3 cells, which does not demonstrate superior tumor selectivity. According to this result, although prodrug nanoassemblies have significantly lower cytotoxicity than CTX sol (no cytotoxicity in 3T3 cells for SGC NPs), please add clarification on why the tumor growth inhibitory effect can be excellent in Fig. 5b.

4.     Please specify the name of each organ in Figure 4b.

5.     In Figure 5, the authors simply presented the tumor volume results. Please add results to visualize differences in tumor size.

6.     The authors analyzed the differences in drug release properties, cytotoxicity results, blood circulation time, and antitumor efficacy of the three NPs by varying disulfide bonds of different lengths, but a discussion about them is lacking. Please add clear explanations to the abstract and conclusion sections.

Author Response

We are truly grateful to your valuable comments and hard work. Based on the comments and suggestions, we summarize a point-by-point response to each of the comments/questions, as detailed in the attachment

Reviewer 2 Report

Three effective anticancer molecules were designed and synthesized by the authors. The data were detailed and recommended for publication.

  • HNMR data needs to be purified, and CNMR need to be provided.

Author Response

(The authors gave the same response as above.)

Round 2

Reviewer 1 Report

The authors have adequately addressed the comments of the previous manuscript. It seems can publish the manuscript without further revision.